# Toward Practical Entity Alignment Method Design: Insights from New Highly Heterogeneous Knowledge Graph Datasets

Submission Id: 2540

## ABSTRACT

The flourishing of knowledge graph (KG) applications has driven the need for entity alignment (EA) across KGs. However, the heterogeneity of practical KGs, characterized by differing scales, structures, and limited overlapping entities, greatly surpasses that of existing EA datasets. This discrepancy highlights an oversimplified heterogeneity in current EA datasets, which obstructs a full understanding of the advancements achieved by recent EA methods.

In this paper, we study the performance of EA methods in practical settings, specifically focusing on the alignment of highly heterogeneous KGs (HHKGs). Firstly, we address the oversimplified heterogeneity settings of current datasets and propose two new HHKG datasets that closely mimic practical EA scenarios. Then, based on these datasets, we conduct extensive experiments to evaluate previous representative EA methods. Our findings reveal that, in aligning HHKGs, valuable structure information can hardly be exploited through message-passing and aggregation mechanisms. This phenomenon leads to inferior performance of existing EA methods, especially those based on GNNs. These findings shed light on the potential problems associated with the conventional application of GNN-based methods as a panacea for all EA datasets. Consequently, in light of these observations and to elucidate what EA methodology is genuinely beneficial in practical scenarios, we undertake an in-depth analysis by implementing a simple but effective approach: Simple-HHEA. This method adaptly integrates entity name, structure, and temporal information to navigate the challenges posed by HHKGs. Our experiment results conclude that the key to the future EA model design in practical lies in their adaptability and efficiency to varying information quality conditions, as well as their capability to capture patterns across HHKGs. The datasets and source code are available at *https://anonymous.4open.science/r/HHEA/*.

## KEYWORDS

Knowledge Graphs, Entity Alignment, Graph Neural Networks

## 1 INTRODUCTION

Knowledge Graphs (KGs) are the most representative ways to store knowledge in the form of connections of entities. With the development of KG and relevant applications (e.g., Question Answering [10], Information Retrieval [22]) in recent years, the need for aligning KGs from different sources has become increasingly important in these fields. Entity Alignment (EA) in KGs, which aims to integrate KGs from different sources based on practical requirements, is a fundamental technique in the field of data integration.

Most KGs derived from different sources are heterogeneous, which brings difficulties when aligning entities. Existing EA studies mainly make efforts to identify and leverage the correlation between heterogeneous KGs from various perspectives (e.g., entity names, structure information, temporal information) through deep

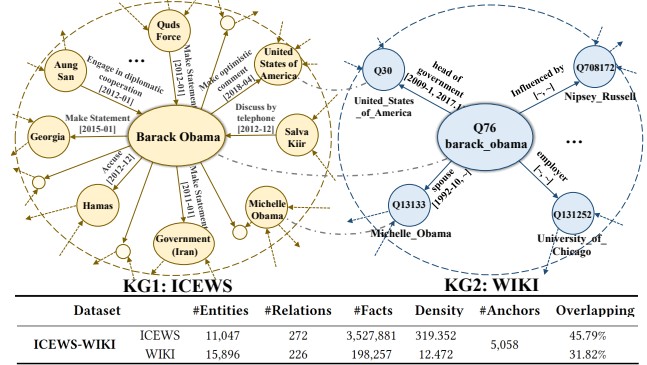

| Dataset | | #Entities | #Relations | #Facts | Density | #Anchors | Overlapping |
|---------|-------|-----------|------------|--------|---------|----------|-------------|
| ICEWS-WIKI | ICEWS | 11,047 | 272 | 3,527,881 | 319.352 | 5,058 | 45.79% |
| | WIKI | 15,896 | 226 | 198,257 | 12.472 | | 31.82% |

**Figure 1: An example of the HHKGs (ICEWS and WIKI) and their statistics. The scale and density of the KGs are very different, and the overlapping ratios (45.79% and 31.82%) indicate that the two KGs are far from the 1-to-1 assumption.**

learning techniques. GNN is one of the most popular techniques for mining graph-structured information. Along with the development of GNN-related techniques, over 70% studies of EA since 2019, according to the statistics [39], have incorporated GNNs into their approaches. Benefiting from the strong ability of GNNs to capture KGs' structure correlations, related methods have achieved remarkable performance on benchmark EA datasets.

How to overcome the heterogeneity and mine the correlations between KGs is the main concern of EA. Existing EA methods evaluate the performance on several widely-used KG datasets, especially cross-lingual KGs (e.g., *DBP15K(EN-FR)*). However, the heterogeneity between KGs is not limited to linguistic differences. Different data sources, scales, structures, and other information (e.g., temporal information ) are more widespread heterogeneities among knowledge graphs, and need to be studied urgently in the EA area. The highly heterogeneous KGs (HHKGs) indicate that the source and target KG are far different from each other (e.g., *General KG-Domain KG*). Figure 1 vividly presents a toy example of HHKGs, in which KGs have different scales, structures, and densities, and the overlapping ratio is exceedingly low. For temporal knowledge graphs (TKGs), the difference in temporal information can also be considered as a kind of heterogeneity. The above characteristics lead to the challenges of EA on HHKGs.

The requirements of practical applications reveal the indispensability of studying HHKG alignment. For example, personal KGs [1, 38] intend to integrate domain knowledge about people with general KG for personalized social recommendations; Geospatial database, which is today at the core of an ever-increasing number of Geographic Information Systems, needs to align entities from multiple knowledge providers [2]. These applications have urgent needs for EA on HHKGs. Unfortunately, most EA methods are evaluated on a few benchmarks, there is a lack of datasets for conducting

research on HHKGs. This one-sided behavior hinders our understanding of the real progress achieved by EA methods, especially GNN-based methods, and results in the limitations of previous EA methods when applied in practical scenarios. In general, a rethinking of EA methods especially GNN-based methods is warranted. The goal of this work is to answer two essential research questions:

- **RQ1**: From the dataset view, what are the existing EA datasets' limitations, and the gaps between them and practical scenarios?
- **RQ2**: From the method view, what is the EA method that we really need in practical applications?

To answer **RQ1**, we conduct a rethinking of the existing EA datasets, and discuss the gap between them and practical scenarios through statistical analysis. Based on the analysis, we sweep the unreasonable assumption (e.g., KGs always satisfy 1-to-1 assumption of entities) and eliminate oversimplified settings (e.g., the excessive similarity of KGs in scale, structure, and other information) of previous datasets and propose two new entity alignment datasets called *ICEWS-WIKI* and *ICEWS-YAGO*.

To answer **RQ2**, we perform empirical evaluations across a wide range of representative EA methods on HHKG datasets. It is noticed that the performances of GNN-based methods, which achieve competitive performances on previous EA datasets, decrease sharply under HHKG conditions. That is to say, GNN-based methods, which primarily rely on leveraging implicit patterns in structure information for EA, do not demonstrate significant advantages over other EA methods when the structure information of HHKGs becomes challenging to utilize. The above phenomena lead us to rethink the so-called progress of existing GNN-based EA methods.

To further investigate the components of GNN that lead to performance degradation, and understand how the characteristics of HHKGs influence these methods, we conduct statistical analysis, ablation studies, and sensitivity studies of GNN-based EA methods on HHKGs, and can summarize that:

(1) The structure information of HHKGs is complicated to utilize through message passing and aggregation mechanisms, resulting in the GNN-based methods' poor performance. The entity name information is not effectively utilized because more noise is aggregated when facing highly heterogeneous graph structures.

(2) It cannot be ignored to leverage structure information, especially when the quality of other types of information cannot be guaranteed. Therefore, we need to design EA models that can mimic the self-loop mechanism of GNNs, and adaptively exploit different types of information between HHKGs.

In light of these observations, and with the intent to discover what makes an EA methodology genuinely beneficial in practical scenarios, we propose a simple but effective method: Simple-HHEA, which can jointly leverage the entity name, structure, and temporal information of HHKGs. We conduct extensive experiments on the proposed datasets and a previous benchmark dataset.

Our experimental insights emphasize three keys for EA models in real-world scenarios: (1) the methods should possess adaptability to various information quality conditions to meet the demands of diverse applications; (2) the methods need the capability to extract clues from highly heterogeneous data for EA; (3) efficiency is also vital, striving for a simple yet effective model that maintains high accuracy while conserving resources.

To summarize, our main contributions are as follows:

(1) We recognize the limitations of previous EA benchmark datasets, and propose a challenging but practical task, i.e., EA on HHKGs. To this end, we have constructed two new datasets to facilitate the examination of EA in practical applications.

(2) We rethink existing GNN-based EA methods. Through adequate analysis and experiments on new datasets, we shed light on the potential issues resulting from oversimplified settings of previous EA datasets to facilitate robust and open EA developments.

(3) We propose an in-depth analysis to provide direction for future EA method improvement through the implementation of a simple but effective approach: Simple-HHEA.

## 2 RETHINKING EXISTING EA RESEARCHES

### 2.1 Limitations of Existing EA Datasets

Upon conducting an extensive review of the field, we noted that a substantial majority of Entity Alignment (EA) studies, specifically more than 90%, utilize a set of established benchmark datasets for evaluation. These include, but are not limited to, *DBP15K* datasets, *DBP-WIKI*, and *WIKI-YAGO* datasets. These datasets have undoubtedly made substantial contributions to the progression of EA research. However, when applied to the study of HHKGs, the intrinsic limitations of these datasets become increasingly prominent. These limitations largely stem from the inherent characteristics of these datasets. Regrettably, these widely used datasets exhibit a degree of simplification. While these feature may making them convenient for conventional EA studies, it significantly deviates from the high heterogeneity and complexity commonly found in real-world KGs. Consequently, these datasets fail to adequately represent the challenges posed by practical applications. This divergence between the characteristics of benchmark datasets and piratical scenarios poses substantial challenges to the alignment of entities within HHKGs.

In this paper, we delve into the prevalent issues inherent in existing EA datasets by conducting rigorous statistical analyses on widely-used datasets, *DBP15K(EN-FR)* and *DBP-WIKI*, proposed by OpenEA [28]. Our choice to examine *DBP15K(EN-FR)* is driven by two factors: (1) its exemplary nature as it shares similar characteristics with other datasets, thereby enhancing the generalizability of our analysis; For example, through the statistical analysis [40], we can observed that *DBP15K(EN-FR)*, other *DBP15K* series dataset, *DBP-WIKI*, and *SRPRS* have the same feature that aligned KGs share the similar scale, density, structure distribution, and overlapping ratios. (2) the datasets are extensively employed within the EA research community, making them the representation of benchmark datasets, and its broad acceptance by the EA community ensures the applicability of our findings.

With *DBP15K(EN-FR)* and *DBP-WIKI* as our focal point, we assess these datasets from three critical dimensions: scale, structure, and overlapping ratios. Through this systematic exploration, we aim to provide valuable insights into the challenges inherent in existing EA datasets and inform future research directions A comprehensive overview of the dataset statistics is presented in *Table* 1.

**Scale.** The statistical analysis of *DBP15K(EN-FR)* and *DBP-WIKI* shows that the KGs exhibit similarities in the number of entities, relations, and facts, indicating that the scales of aligned KGs are same. However, in practical scenarios, KGs from different sources are

Table 1: The detailed statistics of the experiment datasets, "*Structure. Sim.*" denotes the average neighbor structure similarity of entities. "*Temporal.*" denotes whether the dataset contains temporal knowledge information.

| Dataset | | #Entities | #Relations | #Facts | Density | #Anchors | Overlapping | Struc. Sim. | Temporal |
|---|---|---|---|---|---|---|---|---|---|
| **DBP15K(EN-FR)** | EN | 15,000 | 193 | 96,318 | 6.421 | 15,000 | 100% | 63.4% | No |
| | FR | 15,000 | 166 | 80,112 | 5.341 | | 100% | | No |
| **DBP-WIKI** | DBP | 100,000 | 413 | 293,990 | 2.940 | 100,000 | 100% | 74.8% | No |
| | WIKI | 100,000 | 261 | 251,708 | 2.517 | | 100% | | No |
| **ICEWS-WIKI** | ICEWS | 11,047 | 272 | 3,527,881 | 319.352 | 5,058 | 45.79% | 15.4% | Yes |
| | WIKI | 15,896 | 226 | 198,257 | 12.472 | | 31.82% | | Yes |
| **ICEWS-YAGO** | ICEWS | 26,863 | 272 | 4,192,555 | 156.072 | 18,824 | 70.07% | 14.0% | Yes |
| | YAGO | 22,734 | 41 | 107,118 | 4.712 | | 82.80% | | Yes |

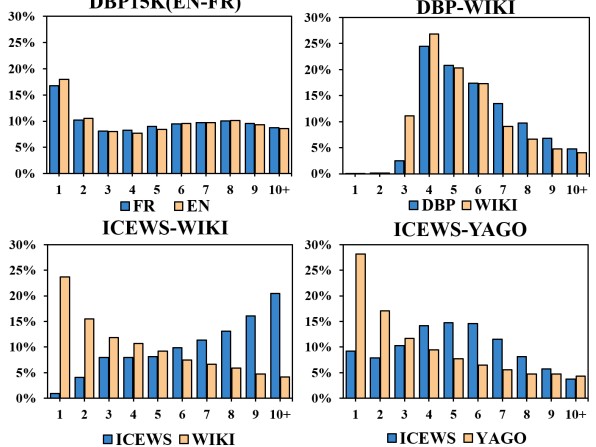

Figure 2: The comparison of degree distribution comparison in DBP15K(EN-FR), DBP-WIKI, and ICEWS-WIKI/YAGO. The X-axis denotes degree and Y-axis represents the entities' ratio.

not always same in scale, and even show significant differences. In recent years, there has been research [28] focused on aligning large-scale KGs. However, there is still a lack of research and datasets modeling KGs that are significantly different in scale, which widely exist. For example, the *ICEWS(05-15)* dataset contains more facts (461,329) compared to *YAGO* (138,056) and *WIKIDATA* (150,079) dataset [11]. Thus, it is impossible to explore the impact of scale differences between KGs in real-world scenarios.

**Structure.** The KGs of *DBP15K(EN-FR)* and *DBP-WIKI* are similar in density, and also have similar degree distributions as shown in Figure 2, which reflect that the KGs are similar in structure.

To further evaluate the neighborhood similarity between KGs, we propose a new metric called *structure similarity*, which is the average similarity of aligned neighbors of aligned entity pairs, and details are illustrated in A.3. The structure similarity of KGs in *DBP15K(EN-FR)* and *DBP-WIKI* reaches 63.4% and 74.8%, indicating that the entity pairs share similar neighborhoods.

The analysis of the structure of *DBP15K(EN-FR)* and *DBP-WIKI* reveals that the structure information of the aligned KGs in previous datasets is similar and easy to leverage, particularly by GNN-based EA methods, such as [7, 17, 32]. These methods exhibit abilities to capture and leverage the structure correlation between KGs, resulting in impressive performances on previous EA datasets. It is worth further investigation to determine if these EA methods, verified on

previous datasets, are still effective in piratical scenarios where the structure information of two KGs is significantly different.

**Overlapping ratio.** We can also notice that the overlapping ratio of *DBP15K(EN-FR)* and *DBP-WIKI* is 100%, which refers to another important characteristic of most EA datasets: *1-to-1 assumption* (each entity in a KG must have a counterpart in the second KG). As discussed in the recent EA benchmark study [15], the 1-to-1 assumption deviates from practical KGs. Especially when the alignment is performed between KGs from different sources (e.g., the global event-related *ICEWS* and the general KG denoted as *WIKI*), the entities that can be aligned only make up a small fraction of the aligned KGs, which is manifested by a low overlapping ratio. For example, some non-political entities in *YAGO* (e.g., football clubs) do not appear in the *ICEWS* dataset.

In conclusion, through the statistical analysis of *DBP15K(EN-FR)*, we find that the previous EA datasets are oversimplified under the unrealistic assumption (i.e., 1-to-1 assumption) and settings (i.e., the same scale, structure), which are easy to leverage but deviate from real-world scenarios. This one-sided behavior hinders our understanding of the real progress achieved by previous EA methods, especially GNN-based methods, and causes the potential limitations of these methods for applications. Moreover, the heterogeneities between real-world KGs are not only language but also the scales, structure, coverage of knowledge, and others, overcoming the heterogeneities and aligning KGs helps complete and enrich them. Therefore, new EA datasets, which can mimic KGs' heterogeneities in more practical scenarios, are urgently needed for EA research.

## 2.2 Towards Practical Datasets

To address the limitations of previous EA datasets, in this paper, we present two new EA datasets called *ICEWS-WIKI* and *ICEWS-YAGO*, which integrate the event knowledge graph derived from the Integrated Crisis Early Warning System (ICEWS) and general KGs (i.e., *WIKIDATA*, *YAGO*). There is a considerable demand to align them in real-world scenarios. *ICEWS* is a representative domain-specific KG, which contains political events with time annotations that embody the temporal interactions between politically related entities. *WIKIDATA* and *YAGO* are two common KGs with extensive general knowledge that can provide background information. Aligning them can provide a more comprehensive view to understand events and serve temporal knowledge reasoning tasks. The detailed construction processes are illustrated in A.4

**Dataset Analysis.** An ideal comparison should include KGs as they are. From the EA research view, the proposed datasets sweep

the oversimplified settings and assumptions (i.e., KGs always satisfy the 1-to-1 assumption, and KGs are similar in scale and structure) of previous datasets, and thus are closer to the real-world scenarios.

In scale, the original two KGs differ significantly in scale. Correspondingly, the datasets maintained the original distribution of the two KGs during sampling, which means that the constructed dataset maintained the scale difference of the KGs.

In structure, we preserved the feature of the original KGs that the density is significantly different. Figure 2 also shows that the two proposed datasets exhibit highly different degree distributions. The *structure similarities* are very low (15.4%, 14.0% of *ICEWS-WIKI/YAGO*) compared with *DBP15K(EN-FR)* (66.4%), referring that the KGs of the new datasets are highly heterogeneous in structure.

In overlapping ratio, the datasets mimic a more common scenario that the KGs come from various sources (i.e., domain-specific KGs and general KGs). The differences in the sources are typically not language but the coverage of knowledge, which result in a small proportion of overlapping entities. The EA task on the new dataset is challenging but also reaps huge fruits once realized. As shown in Table 1, the overlapping ratio of the new dataset is exceedingly low compared with *DBP15K(EN-FR)*, which means that the new datasets do not follow the 1-to-1 assumption.

Moreover, an increasing number of KGs like *YAGO* and *WIKIDATA* contain temporal knowledge, and two recent studies [36, 37] shows that KGs' temporal information is helpful.

Through the statistical analysis, we deem that the proposed datasets are highly heterogeneous, which can mimic the considerable practical EA scenarios, and help us better understand the demands and challenges of real-world EA applications. We expect the proposed datasets to help design better EA models that can deal with more challenging problem instances, and offer a better direction for the EA research community.

The proposed datasets are not only meaningful for EA research but also valuable in the field of KG applications. From the KG application view, *ICEWS*, *WIKIDATA*, and *YAGO* are widely adopted in various KG tasks such as temporal query [16, 33–35], KG completion [14, 21] and question answering [20, 23]. Properly aligning and leveraging these KGs will provide more comprehensive insights for understanding temporal knowledge and benefit downstream tasks.

## 2.3 Rethinking Existing EA Methods

**Translation-based EA Methods.** Translation-based EA approaches [6, 26, 27, 41] are inspired by the TransE mechanism [3], focusing on knowledge triplets $(u, r, v)$. Their scoring functions assess the triplet's validity to optimize knowledge representation. MTransE [6] uses a translation mechanism for entity and relation embedding, leveraging pre-aligned entities for vector space transformation. Both AlignE [27] and BootEA [27] adjust aligned entities within triples to harmonize KG embeddings, with BootEA incorporating bootstrapping to address limited training data.

**GNN-based EA Methods.** In recent years, GNNs have gained popularity in EA tasks for their notable capabilities in modeling both structure and semantic information in KGs [5, 18, 19, 32, 42]. GCN-Align [32] exemplifies GNN-based EA methods, utilizing GCN for unified semantic space entity embedding. Enhancements like RDGCN [7] and Dual-AMN [17] introduced features to optimize

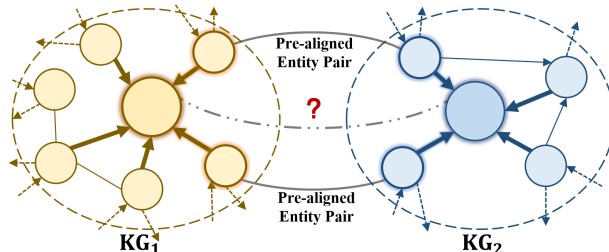

**Figure 3: An example of feature and pre-aligned entity pair label propagation of messaging passing mechanism.**

neighbor similarity modeling. Recent methodologies, e.g., TEA-GNN [36], TREA [37], and STEA [4], have integrated temporal data, underscoring its significance in EA tasks.

Essentially, GNNs can be considered as a framework incorporating four key mechanisms: information messaging (or propagation), attention, aggregation, and self-loop. Each of these plays a fundamental role in GNNs' effective performance in EA tasks. We first give a formulation to the general attention-based GNNs as follows:

$$H_t^{l+1} \leftarrow \underset{\forall (s,r,t) \in Q}{\text{AGG}} (\text{ ATT } (H_s^l, H_r^l, H_t^l,) \cdot \text{ MSG } (H_s^l)) || \text{ SELF } (H_t^l), \quad (1)$$

where $H$ denotes the learned entity embedding, MSG, ATT, AGG, and SELF denote message passing, attention aggregation, and self-loop mechanism, respectively.

Specifically, **Message passing (MSG)** enables nodes to exchange information and assimilate features from neighbors, encapsulating local context crucial for EA tasks, as illustrated in Figure 3. **Attention (ATT)** assigns varied weights to nodes during messaging and aggregation, allowing GNNs to focus on more pertinent nodes, thereby enhancing the extraction of essential structural and semantic KG traits. **Aggregation (AGG)** offers a summarized view of a node's local context by compiling neighbor information into a concise representation. Lastly, the **Self-loop (SELF)** mechanism ensures the preservation of intrinsic node features amidst the learning process, even in the face of intricate intra-graph interactions.

**Other EA Methods.** Certain EA approaches transcend the above bounds. PARIS [25] is a representative non-neural EA method, which aligns entities in KGs using iterative probabilistic techniques, considering entity names and relations. Fualign [31] unifies two KGs based on trained entity pairs, learning a collective representation. BERT-INT [29] addresses KG heterogeneity by solely leveraging side information, outperforming existing datasets. However, its exploration of KG heterogeneity remains limited to conventional datasets, not capturing the nuances of practical HHKGs.

In the context of EA in HHKGs, the efficacy of these methods and mechanisms may be influenced by various factors, such as the level of heterogeneity, the number of overlapping entities, and the quality and quantity of structure and semantic information. This highlights the need for an in-depth analysis in HHKG settings, which will be the focus of our subsequent experimental investigation. This investigation is expected to shed light on the real progress made by EA methods and guide future research toward more effective and robust solutions for real-world EA challenges.

**Table 2: The settings of baselines, and the main experiment results on DBP15K(EN-FR), DBP-WIKI, ICEWS-WIKI, and ICEWS-YAGO. *Bold*: the best result; *Underline*: the runner-up result. "*Struc., Name., Temporal.*" denotes whether methods utilize structure, name, and temporal information, respectively; "*Semi.*" denotes whether methods adopt the semi-supervised strategy; Baselines are separated according to the groups described in Section A.5.2.**

| | Models | Settings | | | | DBP15K(EN-FR) | | | DBP-WIKI | | | ICEWS-WIKI | | | ICEWS-YAGO | | |
|---|---|---|---|---|---|---|---|---|---|---|---|---|---|---|---|---|---|
| | | Struc. | Name. | Temp. | Semi. | Hits@1 | Hits@10 | MRR | Hits@1 | Hits@10 | MRR | Hits@1 | Hits@10 | MRR | Hits@1 | Hits@10 | MRR |
| Trans. | MTransE | | | | | 0.247 | 0.577 | 0.360 | 0.281 | 0.520 | 0.363 | 0.021 | 0.158 | 0.068 | 0.012 | 0.084 | 0.040 |
| | AlignE | ✓ | | | | 0.481 | 0.824 | 0.599 | 0.566 | 0.827 | 0.655 | 0.057 | 0.261 | 0.122 | 0.019 | 0.118 | 0.055 |
| | BootEA | ✓ | | | ✓ | 0.653 | 0.874 | 0.731 | 0.748 | 0.898 | 0.801 | 0.072 | 0.275 | 0.139 | 0.020 | 0.120 | 0.056 |
| GNN | GCN-Align | ✓ | | | | 0.411 | 0.772 | 0.530 | 0.494 | 0.756 | 0.590 | 0.046 | 0.184 | 0.093 | 0.017 | 0.085 | 0.038 |
| | RDGCN | ✓ | ✓ | | | 0.873 | 0.950 | 0.901 | 0.974 | 0.994 | 0.980 | 0.064 | 0.202 | 0.096 | 0.029 | 0.097 | 0.042 |
| | Dual-AMN*(basic)* | ✓ | | | | 0.756 | 0.948 | 0.827 | 0.786 | 0.952 | 0.848 | 0.077 | 0.285 | 0.143 | 0.032 | 0.147 | 0.069 |
| | Dual-AMN*(semi)* | ✓ | | | ✓ | 0.840 | 0.965 | 0.888 | 0.869 | 0.969 | 0.908 | 0.037 | 0.188 | 0.087 | 0.020 | 0.093 | 0.045 |
| | Dual-AMN*(name)* | ✓ | ✓ | | | 0.954 | 0.994 | 0.970 | 0.983 | 0.996 | 0.991 | 0.083 | 0.281 | 0.145 | 0.031 | 0.144 | 0.068 |
| | TEA-GNN | ✓ | | ✓ | | - | - | - | - | - | - | 0.063 | 0.253 | 0.126 | 0.025 | 0.135 | 0.064 |
| | TREA | ✓ | | ✓ | | - | - | - | - | - | - | 0.081 | 0.302 | 0.155 | 0.033 | 0.150 | 0.072 |
| | STEA | ✓ | | ✓ | ✓ | - | - | - | - | - | - | 0.079 | 0.292 | 0.152 | 0.033 | 0.147 | 0.073 |
| Other | BERT | | ✓ | | | 0.937 | 0.985 | 0.956 | 0.941 | 0.980 | 0.963 | 0.546 | 0.687 | 0.596 | 0.749 | 0.845 | 0.784 |
| | FuAlign | ✓ | ✓ | | | 0.936 | 0.988 | 0.955 | 0.980 | 0.991 | 0.986 | 0.257 | 0.570 | 0.361 | 0.326 | 0.604 | 0.423 |
| | BERT-INT | ✓ | ✓ | | | **0.990** | **0.997** | **0.993** | **0.996** | **0.997** | **0.996** | 0.561 | 0.700 | 0.607 | 0.756 | 0.859 | 0.793 |
| | PARIS | ✓ | ✓ | | | 0.902 | - | - | 0.963 | - | - | 0.672 | - | - | 0.687 | - | - |
| Ours | Simple-HHEA | | ✓ | ✓ | | 0.948 | 0.991 | 0.960 | 0.967 | 0.988 | 0.979 | **0.720** | **0.872** | **0.754** | **0.847** | **0.915** | **0.870** |
| | Simple-HHEA⁺ | ✓ | ✓ | ✓ | | 0.959 | 0.995 | 0.972 | 0.975 | 0.991 | 0.988 | 0.639 | 0.812 | 0.697 | 0.749 | 0.864 | 0.775 |

## 3 EXPERIMENTAL STUDY

In this section, we delve into an experimental study to verify the effectiveness of representative EA methods, aiming to shed light on three main questions, which are detailed below:

- **Q1**: What is the effect of existing EA methods on HHKGs?
- **Q2**: Do GNNs really bring performance gain? Which components of GNNs play a key role in EA performance?

In our experiment, we utilized two classic datasets, DBP15K(EN-FR) and DBP-WIKI, and introduced our new datasets: ICEWS-WIKI and ICEWS-YAGO, detailed in Table 1. By reviewing EA methods, we categorized them based on input features, embedding modules, and training strategies, subsequently selecting 13 representative models. These include translation-based methods such as MTransE [6] AlignE [27], and BootEA [27], GNN-based methods like GCN-Align [32], RDGCN [7], TREA [37], TEA-GNN [36], STEA [4], and Dual-AMN [17], and several other methods like PARIS [25], BERT [9] and FuAlign [31]. We adopted two main evaluation metrics: *Hits@k* ($k = 1, 10$) and *Mean Reciprocal Rank (MRR)*. More intricate settings can be found in A.5.

### 3.1 Results and Discussion

To answer **Q1**, we conducted a detailed analysis in four dimensions (datasets, training strategies, embedding modules, and input features). The main experiment results are shown in Table 2.

**From the perspective of datasets**, baselines that generally perform well on *DBP15K(EN-FR)* and *DBP-WIKI* decrease significantly on new datasets, especially GNN-based and translation-based methods. In practice, existing EA methods are difficult to reach the required accuracy performance for applications on HHKGs.

**From the perspective of training strategies**, we found that the semi-supervised methods (i.e., BootEA, Dual-AMN*(semi)*, and STEA) did not improve performance compared to their basic models (i.e., AlignE and Dual-AMN*(basic)*), indicating that the existing semi-supervised strategies are not suitable for HHKGs.

**From the perspective of embedding modules**, neither translation-based methods nor GNN-based methods struggle to capture the correlation between HHKGs, thus the performance of these models is disappointing on new datasets. Especially, the performances of GNN-based methods, which demonstrate the superiority on the DBP15K(EN-FR) dataset, drop sharply on HHKGs. For example, the SOTA GNN-based model Dual-AMN*(basic)* [17], which performs well on the DBP15K datasets, but only achieved MRR of 0.143 and 0.069 on ICEWS-WIKI and ICEWS-YAGO respectively. The huge performance drops of GNN-based methods responses to **Q2**. In previous EA datasets, KGs have similar structure information and thus GNN-based EA methods can easily exploit the structure similarities between entity pairs for alignment purposes. By contrast, a pair of identical entities often have diverse neighborhoods in two HHKGs, resulting in poor performances. BERT and BERT-INT, which mainly leverages side information instead of aggregating neighbors, perform better among others (attains MRR of 0.607 and 0.793). FuAlign, which adopt the translation mechanism, performs poorly on HHKGs. Despite not utilizing neural mechanisms, PARIS performed competitively on HHKGs.

**From the perspective of input features**, the models are capable of utilizing entity name information except Dual-AMN*(name)* outperform the others, indicating to some extent the importance of entity name information in the context of EA models. Specifically, BERT achieves decent results (MRRs of 0.596 and 0.784) on ICEWS-WIKI and ICEWS-YAGO datasets by directly inputting name embeddings generated by BERT into CSLS similarity [8]. However, RDGCN and Dual-AMN*(name)*, which also utilize entity name information, perform badly on HHKG datasets. BERT-INT and FuAlign use entity name information without a GNN framework, notably, they perform much better than existing GNN-based EA models, proving that the aggregation mechanism limits their performance on our HHKG datasets. The utilization of structure information on HHKGs does not bring promising performance gains. Whether structure information is essential for EA on HHKGs still remains to be explored. In terms of temporal information, TREA outperforms

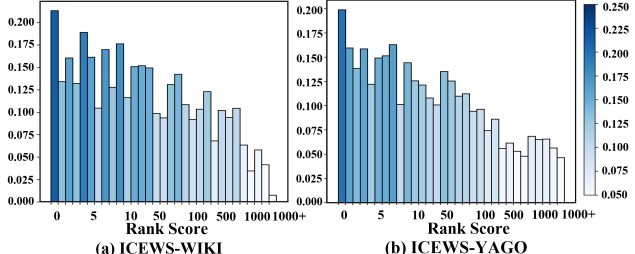

**Figure 4: Case studies of the Dual-AMN*(name)*. The X-axis denotes the performance of the Dual-AMN*(name)*, measured in terms of Rank Score (higher indicates poorer performance). The Y-axis represents the average structure similarity, calculated based on the similarity in the structure of neighboring entities with correct alignment labels in the training set.**

**Table 3: Performance of Dual-AMN without attention (ATT), aggregation (AGG) and self-loop (SELF) of the GNN.**

| Settings | ICEWS-WIKI | | | ICEWS-YAGO | | |
|---|---|---|---|---|---|---|
| | Hits@1 | Hits@10 | MRR | Hits@1 | Hits@10 | MRR |
| Dual-AMN*(basic)* | 0.077 | 0.285 | 0.143 | 0.032 | 0.147 | 0.069 |
| *w/o* ATT | 0.049 | 0.241 | 0.112 | 0.021 | 0.102 | 0.049 |
| *w/o* AGG | 0.000 | 0.001 | 0.001 | 0.000 | 0.001 | 0.001 |
| *w/o* SELF | 0.072 | 0.231 | 0.136 | 0.031 | 0.135 | 0.068 |
| *w/o* ATT, AGG, SELF | 0.000 | 0.001 | 0.001 | 0.000 | 0.001 | 0.001 |
| Dual-AMN*(name)* | 0.083 | 0.281 | 0.145 | 0.031 | 0.144 | 0.068 |
| *w/o* ATT | 0.053 | 0.246 | 0.113 | 0.020 | 0.108 | 0.050 |
| *w/o* AGG | 0.622 | 0.796 | 0.691 | 0.804 | 0.876 | 0.831 |
| *w/o* SELF | 0.082 | 0.288 | 0.151 | 0.031 | 0.140 | 0.066 |
| *w/o* ATT, AGG, SELF | 0.471 | 0.631 | 0.528 | 0.770 | 0.863 | 0.805 |

Dual-AMN with temporal information utilization, proving that temporal information is also valuable in EA tasks when available.

Generally, the efficacy of existing methods is constrained, especially when applied to HHKGs. Specifically, entity name and temporal information play critical roles in alignment, offering a necessary foundation for entity matching across varied KGs. However, under high heterogeneity, the effectiveness of structure information decreases, presenting challenges for traditional EA methods.

### 3.2 GNN-based EA methods on HHKGs

To answer **Q2** and delve deeper into the performance influence of key components of GNNs (i.e., MSG, ATT, AGG, SELF) mentioned in Section 2.3, we further take the SOTA GNN-based method: Dual-AMN as an example, and devised case studies and ablation studies.

**Messaging passing (MSG)**. To explore the influence of MSG, we conduct case studies as shown in Figure 4.

The experiment reveal that as the structure similarity between entities decreases and their neighbors with alignment labels becomes different, the GNN-based Dual-AMN*(name)* model can hardly leverage structure information through message passing for EA. From the perspective of label propagation, the performance of GNNs deteriorates because they struggle to propagate the correct alignment labels from pre-aligned entity pairs to unobserved entities.

We also conduct ablation studies to verify the effectiveness of ATT, AGG, and SELF on HHKGs, as shown in Table 3.

**Attention (ATT)**. It plays a critical role in both scenarios, emphasizing the significance of certain nodes and edges in the graph, thus contributing to more accurate alignment decisions.

**Aggregation (AGG)**. The experiment results underscore a dual role of the AGG, which manifests differently depending on the availability of entity name information for EA. In scenarios where entity name information is absent, the elimination of the aggregation mechanism causes a considerable decline in performance. Conversely, when name information is available, the removal of the aggregation mechanism surprisingly improves the model's performance. This improvement could be attributed to the aggregation mechanism's propensity to introduce noise within HHKGs, which can obscure the valuable cues embedded in the entity name.

**Self-loop (SELF)**. The self-loop mechanism's marginal influence suggests its role might be overshadowed by other components, especially in a highly heterogeneous context. However, after removing AGG, the self-loop mechanism serves a critical function by maintaining each node's individual features, acting as a form of identity preservation and feature selection in the face of HHKGs.

Through this analysis, it becomes evident that major factor causing a drastic decrease in performance is the challenge faced by the message passing and aggregation mechanisms of GNNs on HHKGs. These mechanisms struggle to propagate and aggregate valuable information amidst the influx of unrelated data. This revelation highlights the need for more in-depth analysis and more nuanced, data-aware approaches in model design and applications, especially when dealing with highly heterogeneous settings.

## 4 TOWARD PRACTICAL EA METHODS

Incorporating the above insights, our consideration in this section is to address two critical questions: **What constitutes an effective EA model for applications, and which factor is impactful in practical scenarios?** To answer it, we embark on a comprehensive examination by implementing a Simple but effective Highly Heterogeneous Entity Alignment method, namely Simple-HHEA.

### 4.1 Model Details for Simple-HHEA

**Entity Name Encoder.** We first adopted *BERT* [9] to encode the entity names into initial embeddings. Then we introduced a *feature whitening transformation* proposed by [24] to reduce the dimension of the initial embeddings. The integration of BERT and the whitening layer allows the model to effectively capture the semantic meaning of entities, without requiring additional supervision. Finally, we adopted a learnable linear transformation $W_{\mathcal{T}}$ to get the transformed entity name embeddings $\{\mathbf{h}_n^{name}\}_{n=1}^N$.

**Entity Time Encoder** is designed to verify the power of temporal information, building upon evidence of its effectiveness as demonstrated in the comparative experiments of Dual-AMN and TREA. Simple-HHEA first annotates the time occurrence of each entity according to facts in KGs. Specifically, for our proposed HHKG datasets, the time set $T$ ranges from 1995 to 2021 (in months) of KGs. The encoder can obtain a binary temporal vector for entity $e_n$ denoted as $\mathbf{t}_n = \{\mathbf{t}_n^i\}_{i=1}^{|T|}$, where $\mathbf{t}_n^i = 1$ if facts involving $e_n$ happen at the $i^{th}$ time point, otherwise $\mathbf{t}_n^i = 0$. We adopted *Time2Vec* [12] to obtain the learnable time representation $\mathbf{t2v}(t)[i]$, expressed as:

$$\mathbf{t2v}(t)[i] = \begin{cases} \omega_i t + \varphi_i, & \text{if } i = 0 \\ \cos(\omega_i t + \varphi_i), & \text{if } 1 \leq i \leq k \end{cases}, \quad (2)$$

where $\mathbf{t2v}(t)[i]$ is the $i^{th}$ element of $\mathbf{t2v}(t)$, here we adopt $\cos(\cdot)$ as the activation function to capture the continuity and periodicity of time, $\omega_i$s and $\varphi_i$s are learnable parameters.

Then, the encoder sums the time representations obtained by *Time2Vec* of the entity occurrence time points, and gets entity time embeddings $h^{time}$ through a learnable linear transformation $W_{\mathcal{T}}$. Unlike other methods that strictly correspond to time, Time2Vec can model the cyclical patterns inherent in the temporal data.

**Simple-HHEA$^+$.** To study the use of the structure information in EA between HHKGs, based on basic Simple-HHEA, we introduced an entity structure encoder, which is different from the aggregation mechanism for modeling structure information. To synchronously model the one-hop and multi-hop relations, the encoder employs a biased random walk with a balance between BFS and DFS [31]. Denote $e_n$ as the selected node at $i$ th step of random walks, and PATH$_n = (e_1, r_1, e_2, \ldots, e_{i-1}, r_{i-1}, e_i)$ as the generated path. The probability of an entity being selected is defined as:

$$\mathrm{P}_r\left(e_{i+1} \mid e_i\right) = \begin{cases} \beta, & d\left(e_{i-1}, e_{i+1}\right) = 2 \\ 1 - \beta, & d\left(e_{i-1}, e_{i+1}\right) = 1 \end{cases}, e_{i+1} \in \mathcal{N}_{e_i}{}^-, \quad (3)$$

where $\mathcal{N}_{e_i}{}^-$ denotes entity $e_i$ 's 1-hop neighbors $N_{e_i}$ except $e_{i-1}$; $d\left(e_{i-1}, e_{i+1}\right)$ is the length of the shortest path between $e_{i-1}$ and $e_{i+1}$; $\beta \in (0, 1)$ is a hyper-parameter to make a trade-off between BFS and DFS [31]. Once an entity $e_{i+1}$ is selected, the relation $r_i$ in the triple $(e_i, r_i, e_{i+1}) \in T'$ is selected simultaneously.

Then, the Skip-gram model $SkipGram(\cdot)$ together with a linear transformation $W_{\mathcal{D}}$ are employed to learn entity embeddings $\{dw_n\}_{n=1}^N$ based on the generated random walk paths to capture structure information of KGs. It eliminates the aggregation or message-passing mechanisms and doesn't require supervision.

The linear transformations adopted in each encoder bear similarities to the self-loop mechanism in GNNs, a mechanism that has been validated for its effectiveness in previous experiments. By employing linear transformations, the Simple-HHEA model aims to be adapted to different data situations. Finally, the multi-view embeddings of entities are calculated by concatenating different kinds of embeddings, expressed as:

$$\{h_n^{mul}\}_{n=1}^N = \{[h_n^{name} \otimes h_n^{time} \otimes h_n^{dw}]\}_{n=1}^N,$$

where $\otimes$ denoted the concatenation operation.

We adopt *Margin Ranking Loss* as the loss function for training, and *Cross-domain Similarity Local Scaling (CSLS)* [8] as the distance metric to measure similarities between entity embeddings.

## 4.2 In-depth Analyses of EA on HHKGs

This section aims to uncover the essential EA model needed in practical scenarios through a comprehensive comparison of Simple-HHEA, Simple-HHEA$^+$, and other baseline methods.

**Simple-HHEA vs. baseline methods**. As a critical tool for the in-depth analysis in this section, we will first examine the performance of Simple-HHEA. Table 2 shows the comparison results. Compared with the current SOTA methods: BERT-INT and PARIS, Simple-HHEA and Simple-HHEA$^+$ achieve competitive performances on the previous datasets: *DBP15K(EN-DE)* and *DBP-WIKI*. For the two HHKG datasets, we can observe that Simple-HHEA and Simple-HHEA+ outperform baselines (including BERT-INT) by 15.9% Hits@1 on *ICEWS-WIKI*, and 9.1% Hits@1 on *ICEWS-YAGO*.

**Table 4: The total number of parameters in the training phase of the models compared in the paper.**

| Model | Number of Parameters |
|---|---|
| MTransE | $O((|E| + |R|) \times d)$ |
| AlignE | $O((|E| + |R|) \times d)$ |
| BootEA | $O((|E| + |R|) \times d)$ |
| GCN-Align | $O(|E| \times d + d \times d \times L)$ |
| GCN-Align | $O(|E| \times d + |R| \times d + d \times d \times L)$ |
| TEA-GNN | $O((|E| + |R| \times 2 + |T|) \times d + 3 \times d \times L \times 2)$ |
| TREA | $O((|E| + |R| \times 2 + |T|) \times d + 4 \times d \times L \times 2)$ |
| STEA | $O((|E| + |R|) \times d)$ |
| Simple-HHEA | $O((|T| + 3 \times 2) \times d)$ |

**Table 5: Ablation study of our proposed framework.**

| Model | DBP15K(EN-FR) | | DBP-WIKI | | ICEWS-WIKI | | ICEWS-YAGO | |
|---|---|---|---|---|---|---|---|---|
| | Hits@1 | MRR | Hits@1 | MRR | Hits@1 | MRR | Hits@1 | MRR |
| Simple-HHEA | 0.948 | 0.960 | 0.967 | 0.979 | **0.720** | **0.754** | **0.847** | **0.870** |
| *w/o* Temp | - | - | - | - | 0.701 | 0.748 | 0.829 | 0.857 |
| *w/o* Whitening | 0.923 | 0.942 | 0.930 | 0.956 | 0.632 | 0.683 | 0.786 | 0.820 |
| Simple-HHEA$^+$ | **0.959** | **0.972** | **0.975** | **0.988** | 0.639 | 0.697 | 0.749 | 0.775 |

The introduction of Simple-HHEA aims to guide the design of future EA models. While it is straightforward in design, experimental comparisons affirm that a well-conceived simple model can also achieve commendable results. We also conduct an efficiency analysis as shown in Table 4. It corroborates the efficiency superiority of Simple-HHEA over baselines, which indicates that enhancing model efficiency through a simpler design is also essential in practical, rather than pursuing accuracy without considering complexity.

**Ablation Studies.** We performed ablation studies by ablating Simple-HHEA to analyze how and when each component of it adds benefit to the EA task as shown in Table 5. It is worth noting that the *whitening strategy* contributes significantly (5.8% and 6.1% improvement on Hits@1) to the great performance of our model, and the temporal information also brings the performance improvement (1.9% and 1.8% on Hits@1). By contrast, the introduction of structure information of Simple-HHEA$^+$ degrades the performance.

Questioning whether entity name information always holds value, and if structure information is indeed ineffective on HHKGs, we conducted sensitivity experiments.

**The impact of structure information.** To ponder how structure information affects EA on HHKGs, we selected several SOTA GNN-based methods (which mainly leverage structure information) and Simple-HHEA, and then randomly masked proportions (0% ~80%) of facts in KGs to mimic different graph structure conditions.

As shown in Figure 5, the mask of structure information affects the overall performance of GNN-based EA methods on both datasets with the mask ratio of facts increasing. This phenomenon proves that GNN-based EA methods' performance mainly depends on the structure information, even on HHKGs. The structure information is difficult to exploit in HHKGs due to their structure dissimilarity, while they still learn a part of meaningful patterns of the HHKGs' structural information for EA, which indicates that existing GNN-based EA methods still have much room for improvement by better leveraging structure information in HHKGs.

For Simple-HHEA, with the mask ratio of graph structure gradually increasing, the temporal information is lost, resulting in the performance decreases of Simple-HHEA. It is worth noting that as the mask ratio of structure information increases, the performance

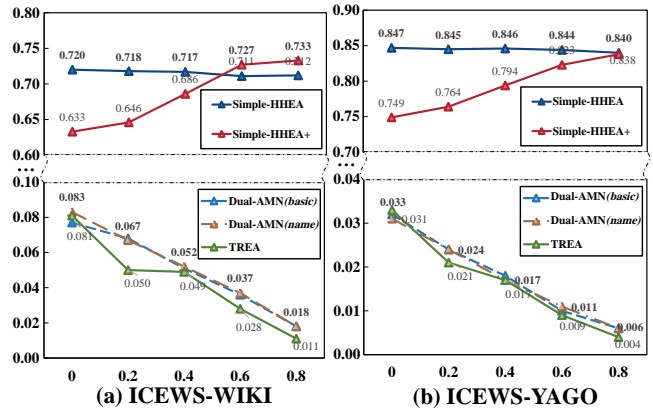

Figure 5: Comparison of different *structure mask ratios* on the ICEWS-WIKI/YAGO. The X-axis denotes the mask proportions of facts, and the Y-axis represents the Hits@1 metric.

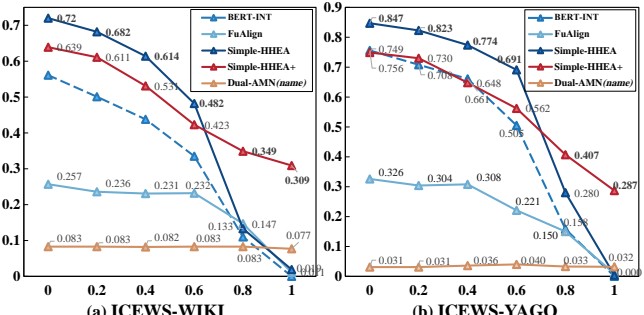

Figure 6: Comparison of different *name mask ratios* on the ICEWS-WIKI/YAGO. The X-axis denotes the mask ratio of entity names, and the Y-axis represents the Hits@1 metric.

of Simple-HHEA$^+$ improves. At a mask ratio of 60%, Simple-HHEA$^+$ even outperforms the basic version. This reflects that in HHKGs, the overly complex and highly heterogeneous structure information will introduce additional noise, and distract the structure correlation mining of Simple-HHEA$^+$ for EA.

In conclusion, the structure information should not be ignored, especially when the quality of other types of information cannot be guaranteed. Future EA methods should consider strategies for pruning and extracting valuable cues from highly heterogeneous structural data for effective EA.

**The impact of entity name information.** The phenomenon observed in Table 2 is that both GNN-based methods (e.g., Dual-AMN*(name)*) and other methods (e.g., BERT-INT, FuAlign, Simple-HHEA) can utilize same entity name information, but their performance varies widely. To explore the role of the entity name information for EA, we randomly mask a proportion (0% ~100%) of entity names, for mimicking different entity name conditions.

As shown in Figure 6, while the mask ratio of entity names gradually increases, the performances of the EA methods which highly rely on entity name information, drop sharply. For BERT-INT, although it uses neighbor information in its mechanism setting, it still needs to filter candidate entities through entity name similarities, so the performance decreases vividly. In addition to entity name information, FuAlign learns structure information through

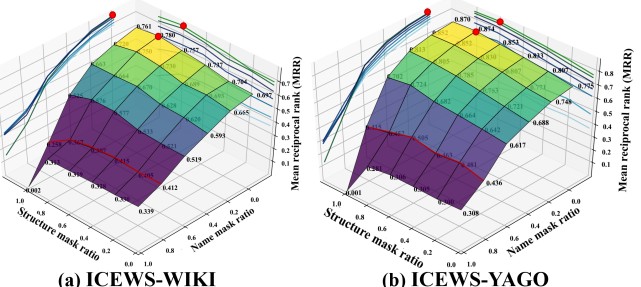

Figure 7: Comparison of different name and structure mask ratios on ICEWS-WIKI/YAGO.

TransE [3]. Therefore, when entity name information is gradually masked, its drop is much smaller than BERT-INT, which indicates that the use of structure information plays a more prominent role in FuAlign, and slows down the performance degradation. Dual-AMN*(name)*'s performance does not change significantly when the entity name information is masked, which shows that Dual-AMN*(name)* can leverage structure information to achieve a stable performance when entity name information is absent.

As the mask ratio of entity name gradually increases, the basic Simple-HHEA also drops sharply like other baselines which highly rely on the quantity of entity name information. Notably, as the quality of entity name information declines, the role of the *entity structure encoder* in Simple-HHEA$^+$ becomes prominent, even surpassing the basic version when the mask ratio is 80%. It indicates that the structure information should not be ignored in EA, especially when the quality of other information can not be guaranteed.

We further designed experiments to explore the performance of Simple-HHEA$^+$ when both name and structure information are changed. As shown in Figure 7, when the quality of entity name information is high (the name mask ratio is less than 80%), the performance of the model decreases when reducing the mask ratio of the structure information; Notably, when the quality of entity name information becomes very low (mask ratio reaches 80%), the above phenomenon is reversed, and the performance increases when introducing entity structure information. This phenomenon reflects that when the quality of different types of information changes, our proposed Simple-HHEA$^+$ can also achieve self-adaptation through simple learnable linear transformations.

In summary, the design of effective EA methods requires the ability to exploit various types of information and adapt to different levels of information quality.

## 5 CONCLUSION AND FUTURE WORK

In this paper, we re-examine the existing benchmark EA datasets, and construct two new datasets, which aim to study EA on HHKGs. Then, we rethink the existing EA methods with extensive experimental studies, and shed light on the limitations resulting from the oversimplified settings of previous EA datasets. To explore what EA methodology is genuinely beneficial in applications, we undertake an in-depth analysis by implementing a simple but effective approach: Simple-HHEA. The extensive experiments show that the success of future EA models in practice hinges on adaptability and efficiency under diverse information quality scenarios, and their ability to discern patterns in HHKGs.

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

# A APPENDIX

## A.1 Preliminary

**Knowledge Graph (KG)** Knowledge graph (KG) $\mathcal{KG} = (\mathcal{E}, \mathcal{R}, Q)$ stores the real-world knowledge in the form of facts $Q$, given a set of entities $\mathcal{E}$ and relations $\mathcal{R}$, the $(e_{head}, r, e_{tail}) \in \mathcal{E} \times \mathcal{R}$ denotes the set of facts $Q$, where $e_{head}, e_{tail} \in \mathcal{E}$ respectively denote the head entity and tail entity, $r \in \mathcal{R}$ denotes the relation. Besides, we also model the temporal information in KGs, given timestamps $\mathcal{T}$, we denote $t \in \mathcal{T}$ as the temporal information of the facts, and each fact is represented in the form of $(e_{head}, r, e_{tail}, t)$.

Based on the basic concept of the KGs, we proposed a concept called highly heterogeneous knowledge graphs (HHKGs). Specifically, it is a relative concept, and indicates that the source KG and the target KG are far different from each other in scale, structure, and overlapping ratios.

**Entity Alignment (EA)** plays a crucial role in the field of knowledge graph research. Given two KGs, $\mathcal{KG}_1 = (\mathcal{E}_1, \mathcal{R}_1, Q_1)$ and $\mathcal{KG}_2 = (\mathcal{E}_2, \mathcal{R}_2, Q_2)$, the goal of EA is to determine the identical entity set $\mathcal{S} = (e_i, e_j) | e_i \in \mathcal{E}_1, e_j \in \mathcal{E}_2$. In this set, each pair $(e_i, e_j)$ represents the same real-world entity but exists in different KGs.

## A.2 Related Works

**Translation-based Entity Alignment.** Translation-based EA approaches [6, 26, 27, 41] are inspired by the TransE mechanism [3], focusing on knowledge triplets $(u, r, v)$. Their scoring functions assess the triplet's validity to optimize knowledge representation. MTransE [6] uses a translation mechanism for entity and relation embedding, leveraging pre-aligned entities for vector space transformation. Both AlignE [27] and BootEA [27] adjust aligned entities within triples to harmonize KG embeddings, with BootEA incorporating bootstrapping to address limited training data.

**GNN-based Entity Alignment.** GNNs, designed to represent graph-structured data using deep learning, have become pivotal in EA tasks [5, 18, 19, 32, 42]. Foundational models like GCN [13] and GAT [30] create entity embeddings by aggregating neighboring entity data, with GAT emphasizing crucial neighbors through attention. GCN-Align [32] exemplifies GNN-based EA methods, utilizing GCN for unified semantic space entity embedding. Enhancements like RDGCN [7] and Dual-AMN [17] introduced features to optimize neighbor similarity modeling. Recent methodologies, e.g., TEA-GNN [36], TREA [37], and STEA [4], have integrated temporal data, underscoring its significance in EA tasks. A fundamental assumption of these methods is the similarity in neighborhood structures across entities, yet its universality remains unexplored.

**Others.** Certain EA approaches transcend the above bounds. PARIS [25] aligns entities in KGs using iterative probabilistic techniques, considering entity names and relations. Fualign [31] unifies two KGs based on trained entity pairs, learning a collective representation. BERT-INT [29] addresses KG heterogeneity by solely leveraging side information, outperforming existing datasets. However, its exploration of KG heterogeneity remains limited to conventional datasets, not capturing the nuances of practical HHKGs.

## A.3 Structure Similarity

To further evaluate the neighborhood similarity between KGs, we propose a new metric called *structure similarity*, which is the average similarity of aligned neighbors of aligned entity pairs, its function is expressed as follows:

$$\text{Structure Similarity}(\mathcal{KG}_1, \mathcal{KG}_2) =$$
$$= \frac{1}{N_S} \sum\nolimits_{(i,j) \in S} \cos\left(\mathbf{A}_{\mathcal{KG}_1}[i] \cdot \mathbf{A}_{\mathcal{KG}_2}[j]\right), \tag{4}$$

where $\mathbf{A}$ denotes KGs' adjacency matrix, $\mathbf{A}[i]$ is the vector of entity $e_i$, which reflects 1-hop neighbors of $e_i$, $S$ is aligned entity pairs.

## A.4 Datasets Construction

We illustrate the detailed construction processes of the datasets. For ease of explanation, we will take *ICEWS-WIKI* as an example in the following discussion, while the process for *ICEWS-YAGO* is similar.

Firstly, we pre-processed the raw event data of *ICEWS* obtained from the official website [1] in the period from 1995 to 2021, and transformed the raw data into the KG format including entities, relations, and facts. Concretely, we pre-processed the entity name in *ICEWS*, and retrieved corresponding entities in Wikidata through the official Wikidata API. Next, we reviewed the candidate entity pairs, and manually filtered unrealistic data, resulting in 20,826 high-quality entity pairs across *ICEWS* and *WIKIDATA*. Then, we sampled the neighbors of the above entity pairs from the original data without enforcing the 1-to-1 assumption. To keep the distribution of the dataset similar to the original KGs, we adopted the *Iterative Degree-based Sampling (IDS)* algorithm [28]. It simultaneously deletes entities in two KGs, under the guidance of the original KGs' degree distribution, until the demanded size is achieved. Finally, the *ICEWS-WIKI* dataset is obtained after sampling. Table 1 summarizes the statistics of the proposed datasets.

## A.5 Detailed Experiment Settings

*A.5.1* **Datasets**. We first adopted two representative EA datasets **DBP15K(EN-FR)** and **DBP-WIKI** [28], which were widely adopted previously. To measure the real effect of the existing EA methods on HHKGs, we also conducted extensive experiments on our proposed datasets: **ICEWS-WIKI** and **ICEWS-YAGO**. The statistics of these selected datasets are summarized in Table 1.

*A.5.2* **Baselines**. After carefully reviewing existing EA studies, we found that most existing EA models can be summarized and classified from the following three perspectives: (1) input features, (2) embedding modules, and (3) training strategies. Specifically, for input features, EA methods mainly adopt entity name information, structure information, and other information (e.g., temporal information). For embedding modules, EA methods mainly adopt translation mechanisms, GNN mechanisms, and others. For training ways, EA methods mainly follow supervised and semi-supervised strategies.

Then, based on the above observations, we selected 11 state-of-the-art EA methods, which cover different input features, embedding modules, and training strategies. The characteristics of these models are annotated in Table 2. We outline the baselines based on the type of embedding module.

---

[1] https://dataverse.harvard.edu/dataverse/icews

**Translation-based methods**: MTransE [6], AlignE [27], and BootEA [27]. BootEA is one of the most competitive translation-based EA methods. Above translation-based methods adopt TransE [3] to learn entity embeddings for EA.

**GNN-based methods**: GCN-Align [32], RDGCN [7], TREA [37], and Dual-AMN [17]. GCN-Align and RDGCN are two GCN-based EA methods, where RDGCN uses entity name information. Dual-AMN is one of the most competitive GAT-based EA methods. We evaluate the performance of Dual-AMN under three conditional settings(*basic/semi/name*), where Dual-AMN*(semi)* adopts the semi-supervised strategy, and Dual-AMN*(name)* utilizes entity name information. TEA-GNN [36], TREA [37] and STEA [4] additionally leverage temporal information for EA.

**Other methods**: BERT [9], FuAlign [31], and BERT-INT [29]. BERT denotes the basic pre-trained language model we adopted for initial entity embedding by using entity name information. FuAlign and BERT-INT are two advanced methods that comprehensively leverage structure and entity name information without GNNs. PARIS [25] is a probabilistic algorithm that iteratively aligns entities in knowledge graphs, even without pre-existing alignments.

*A.5.3* **Evaluation Settings**. In this section, we provide detailed explanations of the model settings, feature initialization, and evaluation metrics.

**Model Settings.** For all baselines of our experiments, we followed their hyper-parameter configurations reported in the original papers except that we keep the hidden dimensions $d = 64$ for fair comparisons. We followed the 3:7 splitting ratio in training/ testing data. All baselines follow the same pre-process procedure to obtain the initial feature as input. In order to eliminate the influence of randomness, all experimental results are performed 10 times, and we take the average as the results. We use PyTorch for developing our work. Our experiments are performed on a CentOS Machine with sixteen 2.1GHz Intel cores and four 24GB TITAN RTX GPUs.

**Feature Initialization.** In our experiments, all EA models, which are capable of modeling entity name information, adopt the same entity name embeddings. Specifically, in *DBP15K(EN-FR)*, we used machine translation systems to get entity names. In *DBP-WIKI*, we converted the QIDs from wikidata into entity names. In *ICEWS-WIKI* and *ICEWS-YAGO*, we used their entity names. After obtaining the textual feature of entities, we adopted the BERT with the whitening transformation [24] to get the initial name embedding.

In addition, for the structure-based EA methods that do not leverage any entity name information, we followed the original settings of these methods to initialize embeddings randomly.

**Evaluation Metrics.** Following the previous work, we adopted two evaluation metrics: (1) Hits@k: the proportion of correctly aligned entities ranked in the top k ($k = 1, 10$) similar to source entities. (2) Mean Reciprocal Rank (MRR): the average of the reciprocal ranks of results. Higher Hits@k and MRR scores indicate better entity alignment performance.

## A.6 Detailed Efficiency Analysis

Table 4 illustrates the parameter number of Simple-HHEA, along with all baselines. The efficiency of an EA method in real-world scenarios is a vital consideration, and this is where Simple-HHEA stands out. With a parameter complexity of $O((|T| + 3 \times 2) \times d)$, it

shows that the parameter count is independent of the size of KGs. Furthermore, Simple-HHEA$^+$ introduces a relation-aware random walk for inductive learning through sampling. This combined with size-independent training parameters underscores its adaptability, demonstrating that effectiveness and efficiency are essential considerations for EA tasks on HHKGs.

## A.7 Future Work

In the future, building upon the analysis of this paper, we summarize several exciting directions:

**More high-quality EA datasets.** The creation of high-quality EA datasets that closely mimic practical scenarios is crucial for advancing the development of EA methods. It would be beneficial to develop more high-quality EA datasets that cover different types of KGs, domains, and heterogeneities. These datasets will provide a comprehensive evaluation of alignment methods and improve understanding of their strengths and weaknesses. They will also serve as a valuable resource for researchers to develop better methods to handle practical EA challenges.

**More advanced methods.** To address the limited ability of previous EA methods to effectively capture the structure information of HHKGs, developing more advanced models should be the priority. This could involve exploring new EA architectures that can better handle highly heterogeneous structures. It may also involve incorporating more sophisticated GNN methods that can better capture the complex structure correlations between HHKGs. Additionally, investigating the leveraging of various information, including entity names, structure, temporal information, and others could overcome the difficulties of highly heterogeneous, thus resulting in more comprehensive and effective EA methods.

**More application scenarios.** It is important to further explore the potential applications of the newly proposed HHKG datasets in other knowledge graph-related tasks such as temporal knowledge graph completion and reasoning. This will provide valuable insights into the versatility of these HHKG datasets and could lead to new advancements in these related applications.

