# OpenReview forum: "Toward Practical Entity Alignment Method Design: Insights from New Highly Heterogeneous Knowledge Graph Datasets"
_ACM.org/TheWebConf/2024/Conference — TheWebConf24 Oral_

### Official Review · Reviewer_ty2S · 2023-11-13

**Novelty:** 7
**Technical Quality:** 6

**Review:**

This paper studied the performance of EA methods in practical settings, specifically focusing on the alignment of highly heterogeneous KGs (HHKGs). It first addressed the oversimplified heterogeneity settings of current datasets and proposed two new HHKG datasets that (as the authors state) can more closely mimic practical real EA scenarios. Based on these datasets, experiments have been done to evaluate previous representative EA methods.

The paper is well written, and the proposed analysis has been conducted by presenting some RQs and systematically addressing them. Which makes the paper easy to follow. I like the use of an “In summary” on several occasions to summarize the results and the technical aspects discussed/presented. The only thing I would suggest is to introduce more practical real examples, just to clarify the discussed concepts.

I found, the objectives of this paper very interesting and valuable. I find the presented results significant and with a significantly relevant impact also to future works focusing on EA analysis.

The authors have also a dedicated and well-presented github repository to include the analysis/results done in this article alongside the data and the scripts to be executed to run the tests.

**Questions:**

From the last section “CONCLUSION AND FUTURE WORK“ I would like to hear more about what are the future and upcoming analysis that could be done. E.g., are there any future plans on how to test more Simple-HHEA?

**Ethics Review Description:**

__

**Reviewer Confidence:**

3: The reviewer is confident but not certain that the evaluation is correct

**Scope:**

4: The work is relevant to the Web and to the track, and is of broad interest to the community

---

### Official Review · Reviewer_x9EZ · 2023-11-18

**Novelty:** 5
**Technical Quality:** 5

**Review:**

The paper presents a well-written exploration of the challenges in entity alignment across highly heterogeneous knowledge graphs (HHKGs). The abstract succinctly introduces the motivation for the study, emphasizing the need for practical entity alignment methods in the face of diverse and complex knowledge graphs. The authors successfully identify a gap in existing research by highlighting the oversimplified heterogeneity in current entity alignment datasets, proposing and introducing two new HHKG datasets that better mimic real-world scenarios. The evaluation of representative entity alignment methods on these new datasets is a strong aspect of the work, providing valuable insights into the limitations of current approaches, particularly those based on graph neural networks (GNNs).

However, the paper could benefit from a clearer focus on the primary contribution. It suggests a threefold approach—new datasets, experiments, and a novel method (Simple-HHEA). To enhance clarity, the title and structure of the paper should be revised to explicitly convey the primary focus. The conclusion, while summarizing the findings, lacks a deeper discussion of the implications of the observed limitations on the broader domain of knowledge graph research. It would be valuable to include recommendations for future research directions and potential applications of the proposed Simple-HHEA method.

Pros:

- The abstract effectively communicates the motivation, methodological approach, and key findings of the study.
- The identification of oversimplified heterogeneity in existing datasets and the proposal of new HHKG datasets contribute to addressing a gap in the current state of entity alignment research.
- The evaluation of existing methods on the new datasets provides valuable insights into the limitations of current approaches, particularly GNN-based methods.

Cons:

- The focus of the contribution (new datasets, experiments, or the Simple-HHEA method) is not clearly delineated in the abstract, and the title and structure of the paper should be adjusted to better reflect this.
- The conclusion is lacking depth, missing an opportunity to discuss broader implications and suggest avenues for future research.
- The significance of the work could be more explicitly stated, emphasizing its potential impact on the design of future entity alignment methods for practical knowledge graph applications.

**Questions:**

1. The paper is titled "Insights…", but I am missing some of the important lessons learned from those insights. What are they?
2. The paper emphasizes the adaptability and efficiency of the Simple-HHEA method to varying information quality conditions. Could you provide more details on how Simple-HHEA achieves adaptability with "linear transformations", especially in handling different types of information (entity name, structure, and temporal)?
3. Additionally, are there specific scenarios or types of knowledge graphs where Simple-HHEA demonstrates superior performance compared to existing methods, and if so, what are those characteristics?

**Ethics Review Description:**

–

**Reviewer Confidence:**

3: The reviewer is confident but not certain that the evaluation is correct

**Scope:**

4: The work is relevant to the Web and to the track, and is of broad interest to the community

---

### Official Review · Reviewer_DS57 · 2023-11-23

**Novelty:** 6
**Technical Quality:** 7

**Review:**

This is a unique and important paper on Entity Alignment. The authors claimed that the current Entity Alightment research does not fit with real-world situations because the popular datasets for EA task are highly homogenous and opposed a new research topic called highly heterogenous Knowledge Graph (HHKG) Alignment. They created two datasets as HHKG to demonstrate how HHG alignment should have different features than those for the current popular datasets. They also created a new EA method called simple-HHEA for HHKG alignment. They performed in-depth analysis by applying several major EA models and their model on the classical datasets and their HHKG datasets. The analysis shows that some functions in typical EA models are not suitable to HHKG datasets and also discussed requirements for future research for HHKG alignment.
The importance of the paper is not as the proposal for a new EA model but the identification of the new problem accompanied by the in-depth analysis on the existing models.

minor comments:
- In Table 4, symbols like E, R, d, T are not defined.

**Questions:**

- In in-depth analysis, the authors focused on GNN-based EA models. But as Table 2 results, BERT-based models are more stable results than them both in the classical datasets and your HHKG datasets. The reviewer wants to know why it is and also want to know how BERT-based models can be adapted for HHKG alignment. In fact, simple-HHEA also uses BERT in it.

**Reviewer Confidence:**

3: The reviewer is confident but not certain that the evaluation is correct

**Scope:**

4: The work is relevant to the Web and to the track, and is of broad interest to the community

---

### Official Review · Reviewer_cXmh · 2023-11-24

**Novelty:** 5
**Technical Quality:** 7

**Review:**

The authors address the topic of entity alignment across multiple, often heterogenous knowledge graphs and how a new GNN-based alignment approach can improve alignment accuracy compared to traditional approaches. By investigating and scrutinizing existing approaches mainly based on reference KGs the authors introduce a new GNN-based method called Simple HHEA that according to their evaluation outperforms traditional approaches but also comes with specific limitations.
The paper is well structured and nice to read. The topic is relevant and contributes to the growing research in the convergence of knowledge graphs and generative techniques. The authors conduct an interesting experimental study whose results are worth a deeper discussion as they provide interesting insights in the reasonable applicability of GNNs for the purpose of EA. The evaluation seems to be robust and again produced interesting results that deserve a deeper discussion, especially as the results leave room for various interpretations, new hypothesis and room for further research.

One formal remark:
- The first half of the paper contains several typos and grammatical errors. Please proof-read the paper before final submission in case of acceptance.

**Questions:**

No questions to the authors.

**Ethics Review Description:**

There are no ethical issues related to this paper.

**Reviewer Confidence:**

1: The reviewer's evaluation is an educated guess

**Scope:**

3: The work is somewhat relevant to the Web and to the track, and is of narrow interest to a sub-community

---

### Decision · Program_Chairs · 2024-01-22

**Decision:**

Accept (Oral)

**Comment:**

This article identifies problems within the area of entity alignment over highly heterogenous Knowledge Graphs, and shows problems with existing techniques.

 All authors agree that this is important and relevant work, which deserves to be accepted.
 We recommend the authors to incorporate the comments and clarifications that arose during the discussions, such as the suggested enhancements to the paper structure and the deeper discussion on what the implications of this work are.